# Phycobilisome light-harvesting efficiency in natural populations of the marine cyanobacteria *Synechococcus* increases with depth

Yuval Kolodny[1,2,6], Yoav Avrahami [3,4,6], Hagit Zer[5], Miguel J. Frada [3,4], Yossi Paltiel [1,2] & Nir Keren [5✉]

Cyanobacteria of the genus *Synechococcus* play a key role as primary producers and drivers of the global carbon cycle in temperate and tropical oceans. *Synechococcus* use phycobilisomes as photosynthetic light-harvesting antennas. These contain phycoerythrin, a pigment-protein complex specialized for absorption of blue light, which penetrates deep into open ocean water. As light declines with depth, *Synechococcus* photo-acclimate by increasing both the density of photosynthetic membranes and the size of the phycobilisomes. This is achieved with the addition of phycoerythrin units, as demonstrated in laboratory studies. In this study, we probed *Synechococcus* populations in an oligotrophic water column habitat at increasing depths. We observed morphological changes and indications for an increase in phycobilin content with increasing depth, in summer stratified *Synechococcus* populations. Such an increase in antenna size is expected to come at the expense of decreased energy transfer efficiency through the antenna, since energy has a longer distance to travel. However, using fluorescence lifetime depth profile measurement approach, which is applied here for the first time, we found that light-harvesting quantum efficiency increased with depth in stratified water column. Calculated phycobilisome fluorescence quantum yields were 3.5% at 70 m and 0.7% at 130 m. Under these conditions, where heat dissipation is expected to be constant, lower fluorescence yields correspond to higher photochemical yields. During winter-mixing conditions, *Synechococcus* present an intermediate state of light harvesting, suggesting an acclimation of cells to the average light regime through the mixing depth (quantum yield of ~2%). Given this photo-acclimation strategy, the primary productivity attributed to marine *Synechococcus* should be reconsidered.

[1] Applied Physics Department, The Hebrew University of Jerusalem, Jerusalem, Israel. [2] The Center for Nanoscience and Nanotechnology, The Hebrew University of Jerusalem, Jerusalem, Israel. [3] The Interuniversity Institute for Marine Sciences in Eilat, Eilat 88103, Israel. [4] Dept. of Ecology, Evolution and Behavior - Alexander Silberman Institute of Life Sciences, Hebrew University of Jerusalem, Jerusalem 91904, Israel. [5] Department of Plant and Environmental Sciences, The Alexander Silberman Institute of Life Sciences, The Hebrew University of Jerusalem, Jerusalem, Israel. [6] These authors contributed equally: Yuval Kolodny, Yoav Avrahami. ✉email: nir.ke@mail.huji.ac.il

Marine photosynthesis by single-celled microorganisms accounts for nearly 50% of global primary productivity[1]. Numerically, the vast majority of primary producers in the oceans are cyanobacteria, the only extant prokaryotic group of oxygenic photoautotrophs. Among these, the two cyanobacterial genera—*Prochlorococcus* and *Synechococcus*—are responsible for a significant fraction of primary production, mainly in open ocean waters in subtropical and tropical settings[2–4]. The basic photosynthetic apparatus in all cyanobacteria consists of two photochemical reaction centers: Photosystem I and Photosystem II. Most cyanobacteria, including *Synechococcus* that are the focus of our study, possess a supramolecular light-harvesting antenna coupled mainly to PSII, the Phycobilisome (PBS). *Prochlorococcus* however, use membrane internal light-harvesting systems[5]. In *Synechococcus*, the PBS contains proteins that bind phycoerythrin chromophores (PE) absorbing blue light (peak at 497 nm), the wavelength that best penetrates seawater[6,7]. Owing to this adaptation, this genus specializes in light harvesting in the deeper ocean[5].

Light regimes through the water column can change dramatically in space and time. Its intensity attenuates exponentially with depth, and its spectrum is narrowed to blue wavelengths. Moreover, the conditions in an open ocean water column vary seasonally[8]. Generally, during summer periods, as the surface warms up and temperature declines monotonically with depth, the water is stratified, and vertical movements of plankton are restrained. Under these conditions, cells inhabiting different water layers acclimate to the available light regime. However, during winter, cooling of the surface drives vertical mixing of the water column. This in turn requires phytoplankton to entrain to a light regime which exposes them to changes on an hour-to-day time scales[9]. Photosynthetic cells deploy acclimation mechanisms to cope with light regime changes, which impacts photosynthetic performance and thus productivity[10–12].

Among the phytoplankton, cyanobacterial *Synechococcus* species are known to exhibit extensive photo-acclimation capacities[13–15]. Known acclimation strategies to low light conditions include increasing both the number and the size of photosynthetic units[16], a term defining the number of antennae chromophores coupled to a photosystem reaction center[17]. *Synechococcus* cells under low light will contain a higher number of thylakoid membranes per cell, higher chlorophyll content, and larger phycobilisomes with additional PE units[18]. The plasticity of the *Synechococcus* is enabled by the position of the PBS antenna in the inter-thylakoid space. However, at the same time, the intermediate chromophore coupling regime determines energy transfer efficiencies that are considered lower than those of thylakoid membrane internal antenna complexes[19].

Recently, we showed that, in response to low light, the *Synechococcus* WH8102 strain can improve its phycobilisomes' light-harvesting efficiency[20]. From a physical point of view, this discovery is surprising, since, with the larger antenna, the absorption cross-section increases but requires the excitation energy to travel longer distances. In land plants, the longer energy migration path was shown to decrease energy transfer rates[21], as expected according to Forster Resonance Energy transfer calculations[22]. However, in *Synechococcus* WH8102 the energy transfer rate through the antenna to the reaction centers increased when grown under lower light. We demonstrated that this is achieved by enhanced coupling between chromophores in the phycobilisome[20].

When light is absorbed in a photosynthetic light-harvesting complex (PBS in the case of *Synechococcus*), the energy has to migrate through the antenna and reach a reaction center, where photochemical energy conversion takes place. There are three competing pathways that light energy can follow: (i) dissipation through heat; (ii) emission as fluorescence (iii) photochemical reactions[23–26]. In the upper water layers, light intensities are high and excess light can be extremely harmful to the cell, due to the generation of reactive oxygen species (ROS)[27]. Photosynthetic organisms use a variety of mechanisms to dissipate excess energy, collectively called non-photochemical quenching (NPQ) mechanisms[28,29]. NPQ levels may vary significantly and therefore influence the heat dissipation rate in surface waters. However, when examining photosynthetic organisms in deeper layers under lower irradiance, heat dissipation is expected to be minimal and constant[30]. In this scenario, changes in the quantum yield of photochemistry ($\Phi_p$) are inversely related to the quantum yield of fluorescence ($\Phi_f$). Comparing the quantum yields of the different processes can be achieved by using fluorescence lifetime measurements. This is a standard method for estimating light-harvesting efficiency in laboratory studies[31]. Using time-correlated single-photon counting (TCSPC) technique to measure fluorescence lifetime in the picosecond time domain, we can quantitatively relate the fluorescence lifetime to the absolute quantum yield of fluorescence[13,14,30] (Eq. 1): $\Phi_f = \frac{\tau}{\tau_n}$, where $\tau_n$ is the intrinsic or natural lifetime of a phycobilisome complex.

Here we examine how phycobilisome light-harvesting efficiencies correlate with depth in native *Synechococcus* populations. We sampled seawater along a depth gradient during different seasons in the Gulf of Aqaba (GoA), using high-resolution fluorescence lifetime and flow cytometry measurements to specifically capture the energy transfer properties of *Synechococcus* PBS. Our study site, the GoA, is located in the northeastern-most section of the Red Sea. During summer (April–September), the oceanographic conditions are markedly stratified and oligotrophic, resembling an open ocean gyre ecosystem[32]. However, during winter (October–March), surface cooling progressively drives convective mixing of the water column, reaching hundreds of meters in depth, depending on how cold the winter is[9,33,34]. In turn, deep mixing leads to the homogenization of plankton across the mixing depth and the entrainment of ample nutrients to the upper photic layer. This results in the formation of major spring blooms, that are uncommon in (sub)tropical oligotrophic ecosystems[35,36]. *Synechococcus* are numerically a major component of microbial plankton in the Gulf, both during the spring bloom and stratification periods, where higher densities can be found along the deep chlorophyll maximum (DCM) around 80–100 m depth[37–39]. Therefore, this is an attractive study site to examine photo-acclimation dynamics of *Synechococcus* populations in situ in the natural environment.

## Results

Recent advances in TCSPC methods have allowed for fluorescence lifetime measurements of photosynthetic communities in situ[23,30,40]. These earlier studies tackled the lifetime of chlorophyll in surface water, an abundant and constitutive pigment in all photosynthetic systems. TCSPC has a distinct advantage over fluorescence intensity-based methods, measuring Fv/Fm for example, as it does not depend on the concentration of the measured sample. Here, we specifically address the light energy conversion carried out by *Synechococcus*, as a function of depth. This is done by pursuing the fluorescence lifetime of *Synechococcus* phycobilisome systems—excitation at 490 nm, directed at the excitation peak of PE; emission between 515 and 675 nm, capturing only phycobilisome fluorescence and minimizing the contribution of chlorophylls (see samples' fluorescence spectra in Supplementary Fig. 1). Chryptophyceae, also contain PE. Yet, these are considerably larger than *Synechococcus*[41] and were removed from the samples by size-filtration. Their concentrations were negligible, as validated by flow cytometry (Supplementary Fig. 2). Seawater samples from different depths were collected over the course of a year. Flow cytometry allowed us to measure

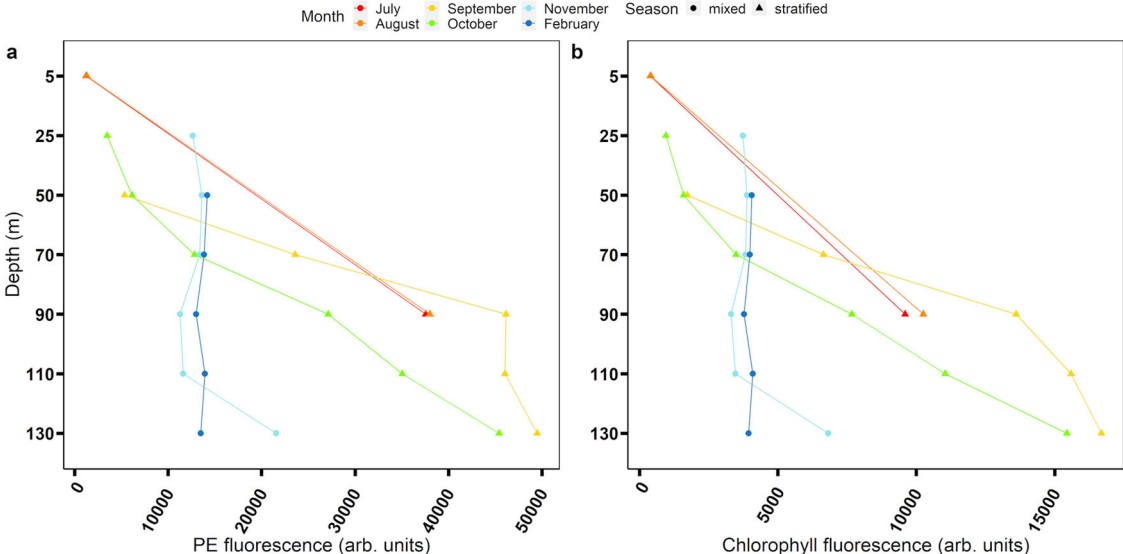

**Fig. 1 Fluorescence intensity of cells identified as *Synechococcus* by flow cytometry.** Flow cytometry measurements were taken at **a** 675–715 nm and **b** 515–545 nm. From July to October, the water column was stratified (triangles). In November, the water column was mixed down to ~100 m, and in February the water column was completely mixed at the measured depths (circles). During stratification, in July and August, flow cytometry data were obtained only from two representative depths (5 and 90 m) and not from every depth at which fluorescence lifetime measurements were performed (presented in Fig. 3). To supplement the data regarding stratification, flow cytometry measurements were also performed in September and October, when the water was still stratified, and are shown in **a**, **b**.

*Synechococcus* abundance, size distribution, and level of pigmentation, using standard gating in the light-scatter and orange fluorescence channels[42]. A CTD (conductivity, temperature, depth) system was used to measure the temperature profile of the water column (see "Methods").

Measurements of the average PE and chlorophyll emission intensities for *Synechococcus* populations, at different depths and seasons, are shown in Fig. 1. These values were calculated from flow cytometry, and refer to single cells (analyzed data sample can be found in Supplementary Figs. 3 and 4). During stratification, the fluorescence emission intensity of both PE and chlorophyll increased with depth, indicating a higher concentration of photosynthetic units in cells inhabiting deeper water layers. This result is consistent with previous studies[14]. In contrast, during mixing, the emission of both PE and chlorophyll was uniform across the mixed layer and was similar to the emission intensities of cells found at about 70 m in the stratified water column. Forward scattering (FSC, Fig. 2) and side scattering (SSC, Supplementary Fig. 5) of *Synechococcus*, measured by flow cytometry, are proportional to the diameter of the interrogated cell (i.e., a proxy for cell size) and to the cell's external and internal complexity, respectively. Our results clearly show that during stratification, cells in the upper ~90 m were similar. Below that depth, their size and complexity gradually increased with depth. During mixing, cells at all depths were identical, resembling the small cells found in the upper layers during stratification.

Based on these data we selected representative times and depths for TCSPC measurements—twice during the stratified period (July and August), and twice during the mixing period (November and February). TCSPC measurements were performed on total plankton populations in seawater samples, while *Synechococcus* were targeted according to their unique spectral features. The average fluorescence lifetime of *Synechococcus* phycobilisomes as a function of depth, at the different months of the year, is shown in Fig. 3.

During summer (July, August), the water column in the GoA was stratified, as evident by the gradual decline of the temperature profiles (Fig. 3a, b and Supplementary Fig. 6). In stratified

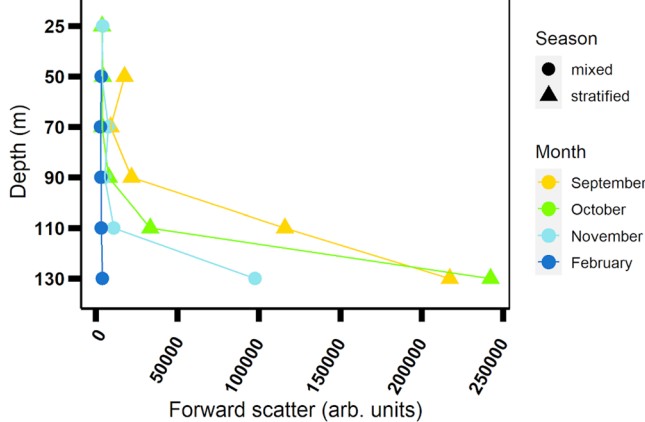

**Fig. 2 Forward scattering (FSC) of *Synechococcus* cells.** Flow cytometry measurements as a function of depth and season. When the water column was stratified (September–October), cells at the deeper layers (below 100 m) were larger.

conditions phytoplankton cells are largely confined to the same water layer for a long period of time acclimating to local light conditions. We found that with increasing depth of the water layer, as light intensity attenuates exponentially, fluorescence lifetime shortens (from 0.38 ns down to 0.07 ns). This indicates a significantly faster energy transfer rate in phycobilisomes under low light conditions, which would eventually contribute to higher quantum yields for photochemistry.

In contrast, during the mixing season (November, February), the temperature was fairly uniform across the mixing depth profile (Fig. 3c, d). Under these oceanographic conditions, *Synechococcus* density is relatively uniform through the mixing layer, as cells are actively mixed (Supplementary Fig. 7). Under such actively mixing conditions, the light intensity perceived by the cells varies significantly over short time periods. In November, the mixing depth reached down to ~100 m (Fig. 3c).

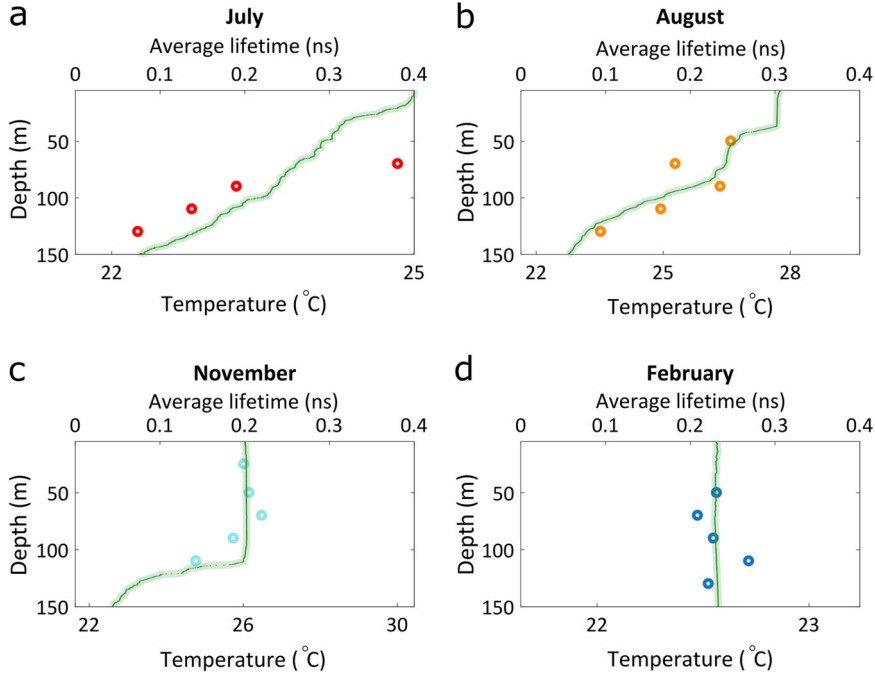

**Fig. 3 Average fluorescence decay time of Phycobilisomes.** Fluorescence decay time of Phycobilisomes (circles) during stratified (**a**, **b**) and mixed (**c**, **d**) water column. Temperature gradients (continuous green lines) indicate the state of stratification/mixing of the water column. During summer (July–August), while the water column was stratified, the lifetime decreased with depth. In a mixed water body (at November down to ~100 m, at February down to 280 m), the Phycobilisomes lifetime was constant, around 0.2 ns.

The average lifetime of all samples down to that depth was around 0.2 ns, while only in the deepest sample, taken from below the mixed layer (110 m) where light levels are very low, cells exhibited a shorter average lifetime of 0.14 ns. In February, when the water was mixed down to 280 m, the lifetime of all samples was constant, around 0.23 ns (Fig. 3d). This is approximately the lifetime found in samples from a depth of 70 m during summer stratification, which is roughly where the deep-chlorophyll-maximum (DCM) is located during summer. The DCM depth at each cruise can be seen in the chlorophyll fluorescence profiles measured by CTD, which appear in Supplementary Fig. 8.

Fluorescence lifetime can be used to evaluate energy conversion efficiencies. To do so, an estimation of the natural lifetime of a relevant phycobilisome assembly that is detached from photosystems is required. For this purpose, we isolated phycobilisomes from *Synechococcus* WH8102 cultures. The integrity of the isolated PBS fraction used for the measurements was evaluated by its fluorescence spectra (Supplementary Fig. 9a). Following the methodology presented by Brody and Rabinowitch for chlorophyll[43], we first evaluated the fluorescence quantum yield $\Phi_f$ through intensity measurements using an integrating sphere spectrometer (Supplementary Fig. 9b). Intact isolated PBS $\Phi_f$ was 0.2. As mentioned, the relation between $\Phi_f$ and lifetime is (Eq. 1): $\Phi_f = \frac{\tau}{\tau_n}$ where $\tau$ is the measured lifetime and $\tau_n$ is the natural lifetime. These isolated Phycobilisomes exhibited an average lifetime of 2.01–2.28 ns and therefore $\tau_n$ is in the range of 10–11.5 ns. Taking the shortest and longest lifetimes measured in the open ocean during stratification (0.38 ns and 0.07 ns at 70 m and 130 m, respectively), gives the following quantum yields of fluorescence: 3.3–3.8% at 70 m and 0.6–0.7% at 130 m. For mixing conditions $\Phi_f = 2$–2.3% throughout the depth gradient. Two observations are important with regard to this calculation. The first is that $\Phi_f$ of PBS systems is lower than those measured by Rabinowitch and coworkers for chlorophyll. This is to be expected when comparing a single chromophore to a network of hundreds of intermediately coupled chromophores. The second is

that under the light intensities at 70 m or below, where NPQ is not expected, $\Phi_f$ is inversely correlated to $\Phi_p$.

## Discussion

Our results show that the fluorescence lifetime of native *Synechococcus* phycobilisomes varies with depth, and follows a clear trend, which correlates to the conditions in the water column. In a stratified water column, when *Synechococcus* remain at a certain depth for sufficient time to acclimate, the physiological state of the cells at each depth is determined by the ambient light radiation. As depth increases, their size and cellular complexity increase in response to light limiting conditions; the number of photosynthetic units and pigment content increases; yet their fluorescence lifetime becomes shorter. Hence, during stratification, the quantum efficiency of light-harvesting increases as light availability decreases.

During mixing, cells are continuously exposed to varying light regimes, requiring them to optimize their photosynthetic machinery to the average available light intensity perceived. Therefore, sampling during mixing served as a natural control experiment for our stratified water column results. Indeed, lifetime was found to be uniform across the mixing depth. Beyond contrasting the results obtained in stratified water, these results show how *Synechococcus* phycobilisome systems cope with the challenges imposed by mixing. Since light intensities change on a short (hours to days) time scale, they cannot optimize to a specific light regime and adopt a likely average state that can serve light harvesting across the mixed layer. This is an "intermediate state," which resembles the state optimized for a depth of ~70 m during the stratified season.

A comparison of photo-acclimation mechanisms between natural populations of *Synechococcus* from the stratified water column, in this study, and a previous study done on light acclimated marine *Synechococcus* strains grown in culture, show similarities. In both cases, PBS fluorescence lifetime was shorter when light intensity was low. Under comparable light intensities, natural populations showed longer fluorescence lifetimes (Fig. 3a, b). For example, based on CTD data, in

the August dataset (during stratification), under light intensities of 137 µmol photons m$^{-2}$ s$^{-1}$ and 6 µmol photons m$^{-2}$ s$^{-1}$, fluorescence lifetimes values were 0.25 and 0.17 ns, accordingly. Under similar illumination conditions in the lab, PBS fluorescence lifetimes were 0.15 (at 150 µmol photons m$^{-2}$ s$^{-1}$) and 0.1 (at 10 µmol photons m$^{-2}$ s$^{-1}$)[20]. In both cases, *Synechococcus* acclimated to lower light intensities exhibited shorter lifetimes. The difference in values may be attributed to the diverse population of *Synechococcus* strains in the GoA[39], compared to the axenic WH8102 strain grown in culture. In addition, changes in the spectral composition of single-cell fluorescence measured by flow cytometry reported here are comparable to those reported by Six and coworkers[44]. The photo-acclimation response of the PBS of natural *Synechococcus* populations from the mixed water column are therefore in contrast to both natural stratified *Synechococcus* and laboratory light acclimated *Synechococcus* strains. Going beyond marine *Synechococcus*, studies of freshwater *Synechocystis* further demonstrate the plasticity of PBS fluorescence properties and indicate lifetimes in the range observed here[45–47].

Additional photo-acclimation responses of *Synechococcus* cells to the increasing depth, are the increase in cell size and in chlorophyll and phycobilin content. Note that the emission intensity of a pigment or chromophore cannot be used to directly quantify its content and to determine the photosynthetic unit size. Yet, previous studies showed that the phycobilisome size increases with increasing depth and decreasing light intensity[44,48]. Forward and side scatter of *Synechococcus* showed a depth profile similar to fluorescence lifetime, with a distinct difference between cells residing in the shallow layers and the deeper layers during stratification, and uniform properties across the water column during mixing (Fig. 2 and Supplementary Fig. 5).

Coordinated dynamics of cell size, photosynthetic unit content, and PBS fluorescence lifetime were previously shown in a laboratory study[20]. Their co-occurrence in our stratified field samples supports the interpretation of the fluorescence lifetime measurements. During mixing fluorescence lifetime had an "intermediate state", and concurrently forward scatter and side scatter values were low—similar to cells in the shallower layer during stratification.

The ability to manipulate energy transfer efficiency in the phycobilisome is not trivial. Previously, it was known that in response to low light, organisms increase their light-harvesting antennae size[21,49,50], therefore increasing the absorption cross-section (absorbing light from a larger surface). However, in a larger antenna, excitation energy must travel a longer distance to reach the reaction centers in the photosystems[21]. In PBS, where pigment-pigment distances are larger than in plants, the effect of a bigger cross-section is expected to be larger. Indeed, Semi-classical dipole-dipole interaction models of phycobilisome, using FRET (Forster resonance energy transfer), which assume one-dimensional phycobilisome rods, predict such an outcome. A longer antenna rod will lower the energy transfer rate[22]. Thus, the fact that the energy transfer rate increases in the larger phycobilisome systems of low light acclimated cells is surprising in view of these classical models. However, these results fit with our previous laboratory study using *Synechococcus* WH8102 cultures[20], where the enhanced coupling was induced under low blue light conditions. The improved energy transfer rate was shown to be the result of enhanced coupling between the chromophores of PE: Phycourobilin and Phycoerythrobilin. Based on these results, it was suggested that the mechanism is either not purely classical, or that it involves an overlooked inter-rod transfer pathway, possibly due to the higher density of phycobilisome rods under low light conditions. The shift from a predominantly single rod one-dimensional energy transfer to coupled rods that allow multidimensional energy transfer may lead to increased efficiency[51].

In principle, a complete picture of the fate of the absorbed energy can be generated from $\Phi_f$ and $\Phi_p$ values (Eq. 2): $\Phi_T = 1 - \Phi_f - \Phi_p$[23]. $\Phi_f$ was calculated from lifetime measurements. $\Phi_p$ is often estimated from variable fluorescence measurements as (Eq. 3): $\frac{F_v}{F_m} = \Phi_p$. However, in cyanobacteria dark $F_m$ values are low[52] and $F_0$ values are high due to a contribution of the tail of PBS fluorescence in the chlorophyll measurement channel[53]. Nevertheless, keeping these limitations in mind we can provide an example of how such a calculation can provide insight. We can use $\frac{F_v}{F_m}$ values measured for *Synechococcus* WH8102[20], using DCMU to get a more accurate reading of $F_m$. Cultures acclimated to medium or low light conditions, which correlate in the GoA, in summer, to ~60 m and ~120 m. $\tau_{avg}$ values measured from these depths are 0.17 and 0.38, respectively. Based on these values, the calculated quantum yield of thermal dissipation will be ~80% for 60 m and ~60% for ~120 m. These values are in the range reported by Falkowski and coworkers[23] based on chlorophyll lifetime measurements.

Over the past decades, variable chlorophyll fluorescence has been the most sensitive, nondestructive signal detectable in the upper ocean that reflects instantaneous phytoplankton photophysiology[54,55]. It has been used to estimate the biomass and physiological status of phytoplankton and has fundamentally changed the interpretation of the biological responses to ocean physics[24]. However, to obtain a complete picture of the energy budget in photosynthetic processes, two of the three competing pathways of absorbed energy (photochemistry, fluorescence, and heat) must be measured. Picosecond fluorescence lifetime measurements can complement variable fluorescence techniques and provide a complete understanding of the fate of absorbed energy. It is also crucial for the development of algorithms for remote sensing techniques (i.e., chlorophyll fluorescence measured by satellites) which are often used to estimate spatial patterns of marine primary production[56–58]. Such algorithms depend on the comparison with accurate in situ measurements of quantum yields[24]. Here, we demonstrate the variability in fluorescence quantum yields as a function of depth, highlighting the importance of a depth-profile fluorescence lifetime approach. To reliably estimate the cyanobacterial integrated contribution to photosynthetic activity in the ocean, their dynamics along the water column should be considered. Since our results indicate an increase in efficiency as a function of depth, which was not considered previously, it may suggest an underestimation of *Synechococcus* productivity by current models[59,60]. With the increase of ocean stratification over the past decades and a similar trend which is expected for the 21st century[61,62], incorporating our results into future models may be beneficial for obtaining more precise estimations, accounting for quantum yield changes in response to the water column ambient illumination conditions.

## Methods

**Water sampling**. Water was sampled at "station A" (29.5° N; 34.95° E; see details in ref. [36]), an open sea station in the GoA during 2020 and 2021 (Fig. 1). Depths were chosen in order to capture the different states of the water column during summer: the stratified layer, the DCM, and below the DCM. Same depths were followed during winter mixing. An additional depth of 25 m was added during two cruises in October-November, which further characterized the shallower communities. During July and August, samples for flow cytometry were collected from two depths only, which represented stratified shallow 5 m and the DCM. From each depth, 5 L were sampled and filtered through a 5 µm plankton net. Two samples from each depth were taken for flow cytometry measurements, and the rest were concentrated by filtering through 0.2 µm polycarbonate filters and suspending the phytoplankton in 3 ml seawater from the same depth. Fluorescence lifetime was measured after dark adaptation of 4 h, during which the samples were kept at room temperature.

**Environmental data**. Sea-Bird SBE 19 CTD (Sea-Bird Scientific, Bellevue, WA, USA) was deployed at each cruise to 500 m and recorded data of pressure

(depth), temperature, salinity, fluorescence, oxygen, and photosynthetically active radiation (PAR).

**Flow cytometry**. Duplicates of 4 mL from each depth and an additional 4 mL blank (0.22 µm filtered-seawater) for total phytoplankton counts were fixed with 0.25% Glutaraldehyde and 0.01% poloxamer. Tubes were incubated for 30 min in dark, 4 °C, followed by flash freeze with liquid N2, and stored at −80 °C. Analysis was performed by Attune NxT flow cytometer (Thermo Fisher Scientific, Bishop Meadow, Loughborough, UK) by 488 nm laser excitation for 5 min, at a rate of 100 µL min$^{-1}$. Emission was examined at 574/26 (peak/ half bandwidth) to detect orange fluorescence of PE, and 695/40 to detect red fluorescence of chlorophyll a. Identification and gating of *Synechococcus* followed a protocol by Marie et al. (1997).

**Fluorescence lifetime**. TCSPC technique was used to measure fluorescence decay lifetime at room temperature, in a self-built setup[20,63]. Excitation was performed with a Fianium WhiteLase SC-400 supercontinuum laser (NKT Photonics, Birkerod, Denmark), monochromatized at 490 nm, aimed at the excitation peak of PE. The repetition rate was 10 MHz. Emission was collected using MPD PD=100-CTE-FC photon counter and PicoHarp300, at the spectral window of phycobilisomes: 515–675 nm, using bandpass filters. The average lifetime for each sample was calculated following deconvolution with measured instrument response function (IRF), using a two-exponential decay model (Eq. 4): ($\tau_{avg} = \frac{\sum_{i=1}^{2} a_i \tau_i}{\sum_{i=1}^{2} a_i}$), with the Fluorescence Decay Analysis Software 1.4, FluorTools, www.fluortools.com. Raw data examples can be found in Supplementary Fig. 10.

**Phycobilisome isolation procedure**. One liter *Synechococcus* WH8102 cultures were harvested by centrifugation at 12,000 × *g* for 7 min. The cells were then resuspended in 0.8 M Phosphate buffer at pH 7. Cells were broken using French Press (20,000 PSI, twice). The homogenate was centrifuged for 2 min at 1150 × *g*, at 4 °C. The supernatant was collected in a new test tube and was centrifuged for 45 min at 18,500 × *g* at 4 °C. The Pellet was resuspended in 0.8 M Phosphate pH 7 and triton X100 was added to give a final concentration of 2% (W/V). The sample was dark incubated for 1 hr at room temp. This was followed by centrifugation for 2.5 h using 147,000 × *g* at 4 °C. The pellet was then resuspended in 0.8 M phosphate buffer pH 7 and loaded on a sucrose gradient (0.25–2 M), centrifuged overnight using SW41 rotor at 40,000 rpm. The bands were collected, and intact phycobilisome bands were identified according to fluorescence spectra.

**Reporting summary**. Further information on research design is available in the Nature Research Reporting Summary linked to this article.

## Data availability

All original data can be obtained from the corresponding author upon request. The raw data used to generate the graphs is shown in Supplementary Data 1.

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

## Acknowledgements

We gratefully thank Inbal Ayalon and the crew of the RV Sam Rothberg from the Interuniversity Institute for Marine Sciences in Eilat and the Israel National Monitoring Program (NMP) for logistic help during cruises. This study was supported by the Israel Science Foundation grant 1182/19 and the Zelman Cowen Academic Initiatives. Work by Y.A. and M.J.F. was supported by the Israel Science Foundation grant no. 2921/20 attributed to M.J.F. A research grant from the Interuniversity Institute for Marine Sciences in Eilat supported Y.K.

## Author contributions

Y.K. and Y.A. contributed equally to the work; Y.K. conceived the work; Y.K. and Y.A. collected the data; Y.K., Y.A., and H.Z. processed the samples; Y.K. and Y.A. led the writing of the manuscript. Y.K., Y.A., H.Z., M.J.F., Y.P., and N.K. analyzed the data and contributed critically to the writing process.

## Competing interests

The authors declare no competing interests.
