## [Peer Review File · Communications Biology]

Reviewers' comments:

Reviewer #1 (Remarks to the Author):

This is a nice example of the collaboration between oceanography and biophysics bridged by photosynthesis research. In this manuscript, the authors showed that phycobilisome of cyanobacterial cells in deeper water had faster decay of fluorescence, suggesting more efficient energy transfer to reaction center. This is interesting as well as only novel finding of this paper. In that sense, Figure 3 alone is essential. This type of concise paper may not be highly evaluated in some types of journals, but the reviewer considers conciseness itself may not be a problem because the fact that the efficiency of energy transfer can be regulated in response to light environments in open sea is very important from the viewpoint of photosynthesis research as well as of ecology. The following points should be considered to improve the manuscript.

1. Discussion is quite qualitative. Even though this is the first report on the depth-dependency of antenna regulation of cyanobacteria in natural water, there are several reports on this point for cultured cells, including the works of authors themselves. Please compare the values of fluorescence decay, fluorescence yield or yield of photosynthesis with the values in the past studies of cyanobacteria and other photosynthetic organisms and discuss them more quantitatively.
2. For the comparison mentioned above, understanding of the precise experimental condition is essential. The current explanation of the method of Time-Correlated-Single-Photon-Counting refers to the reference 56, but the explanation in the paper is very short and insufficient. The explanation in the reference 19 is more thorough. If the experimental condition is the same, refer to reference 19 and remove reference 56. Furthermore, the fluorescence lifetime must differ at different temperatures. Please specify the temperature during lifetime measurements. And if it is low temperature, please discuss about the physiological relevance of the interpretation.
3. To calculate absolute quantum yield of fluorescence, the intrinsic lifetime of a phycobilisome complex is essential. For this purpose, the authors prepare the complex biochemically, but it is hard to judge if the obtained intrinsic lifetime is artificially disturbed or not. If possible, it would be nice to have some supporting data that assure the intactness of the complex. Alternatively, it may be possible to strengthen the reliability of the obtained value by comparing it with those reported for similar preparations in the past.
4. It is interesting to know the "complete picture of the fate of the absorbed energy" but it requires the precise determination of F_v/F_m as well as that of fluorescence yield. In the case of cyanobacteria, determination of correct F_v/F_m is not easy (see e.g. Photosynthesis Research, 133, 63-73). Since the authors seems to use DCMU for the determination of F_m , state transition may not have much interfered with the obtained results. The contribution of phycobilisome and PSI chlorophyll to F_o , however, must have led to underestimation of F_v/F_m . The possible effect of such problem should be discussed.

Minor points

5. Please distinguish "pigment" and "chromophore". Linear tetrapyrrole in phycobilin should be chromophore, not pigment, since it covalently binds to phycobiliprotein.
6. Please explain why cyanobacteria used inefficient phycobilisome in upper water, if it is possible to make it efficient by "enhanced coupling between pigments in the phycobilisome". Light absorbance may not be limiting step in photosynthesis in upper water, but, even so, smaller antenna with higher efficiency must be advantageous.
7. First line of the last paragraph of the Introduction section: "in in" -> "in"
8. First paragraph of the Results section: Please first spell out "CTD" or explain it.
9. Second paragraph of the Results section: "chlorophyll increased with depth, indicating a higher

concentration of photosynthetic units in cells inhabiting deeper water" is based on the fact that "chlorophyll" is estimated as "per cell" when determined by flow cytometry measurements. This may not be obvious for broad readers of this manuscript.

10. Forth paragraph of the Discussion section: "(Kolodny et al., 2020b)" should be reference 19.

11. Fifth paragraph of the Discussion section: "mechanism that reacts to the radiation regime" -> "mechanism that respond to the radiation regime"?

12. Fifth paragraph of the Discussion section: "possible due to the higher density of phycobilisome rods" -> "possibly due to the higher density of phycobilisome rods"?

13. Figures 1,2,S1: It may be better to connect symbols of the same season with lines. In the present figures, some symbols are overlapped with one another, and depth-dependency is not so clear.

14. Figure 3: Do not use arbitrary axis for temperature (X axis) for different panels. It makes the comparison between the panels difficult.

15. Figure S2: Why the measuring depth is different for different seasons? Please add some explanation in the text.

Reviewer #2 (Remarks to the Author):

The manuscript of Kolodny et al. presents novel and interesting data on the photophysiological characteristics of in-situ cyanobacterial (*Synechococcus*) populations in the ocean. The study builds on previous culture work by the same group (Kolodny et al. 2020 FEBS J), and effectively amounts to an in-situ confirmation of the key result in this previous work, namely that *Synechococcus* exhibits a decreased fluorescence lifetime within the phycobilisome and hence apparently an associated increased transfer efficiency under decreased light conditions. In general the manuscript is well written and presented and the results are likely to be of interest, although predominantly probably within the field rather than of wider interest. There were some aspects of the work which I would like to see improved before recommending publication. In particular that were a number of areas where more detail is required in the methods. This is particularly important as the TCSPC measurements are still some of the first reported from natural oceanic populations of phytoplankton and, to my knowledge, the first depth profiles. There are also a few places in the results where I think the authors have reported information in error. These and other more minor points are outlined further below.

Specific comments:

Line 41: I don't follow this. The measured lifetimes are shorter at depth, so shouldn't the fluorescence quantum yield be smaller (i.e. see equation line 97).? I think this is probably a typo and the authors have these values the wrong way around? i.e. the fluorescence quantum yield is 18% at 70m and 3.5% at 130m? See also Line 224 (and associated comment below).

Line 50: It is perhaps worth noting that *Prochlorococcus* (which is numerically the dominant prokaryotic primary producer in many open ocean systems) does not have a phycobilisome. As a broader contextual question, presumably the phenomenon described is specific to the phycobilisome?

Line 57: '...light harvesting in the deeper ocean'

Line 79: '... when grown under lower light.'

Line 150: '...phytoplankton cells are...'

Line 174: decapitalise 'The'

Line 203: Either 'must travel a longer distance' or 'must travel longer distances'

Line 210: 'fit with our previous laboratory study'

Line 224: See previous comment. Taking a value of the intrinsic lifetime of around 2.1 I calculate fluorescence quantum yields of 18% and 3.3% at 70 and 130m respectively.

Lines 225-226: It is unfortunate if, as appears, there is no direct measurement of Fv/Fm from the in situ samples? I would be cautious in the use of these culture data in comparison to the in situ measurements. At the very least, all of the potential caveats in this calculation have to be clearly outlined to the reader. For example, nutrient status will likely be different for the insitu population versus the culture and may have an influence.

Line 247: Not sure this statement can be fully defended as prior measurements of Fv/Fm with depth already indicated that photochemical efficiencies increased at lower light intensities? Also, Line 248, do these models actually make explicit assumptions about the photochemical (or transfer) efficiencies?

Line 267: pre-concentration step. Did you measure Fv/Fm before and after pre-concentration in order to establish whether there was any change? If not did you have any other way of establishing whether the pre-concentration step might have influenced the measurements?

Line 269: how were the samples stored in the dark? E.g. what temperature?

Line 283: assume '574/26' is peak and half bandwidth of emission band? Please state this.

Line 287: suggest 'excitation was performed at 490 nm'

Line 288: was all the signal between 515-675 nm averaged? As indicated above, I would like to see more example data presented, including emission spectra if available and decay curves (see e.g. Figure 4b & c in the cited reference 56).

Line 289: define 'IRF' (i.e. Instrument Response Function)

Figure 1 caption: 'two representative depths...'

Reviewer #3 (Remarks to the Author):

1 The first fatal problem of this paper is the simple experiment design doesn't not support the proposed generalized photosynthetic energy usage model. Only fluorescence lifetime data was collected that is related photosynthesis, and even this lifetime was set for measure phycobilisome lifetime, not the photosynthesis.

2 In the study, fluorescence lifetime from the water sample were purposefully set at the excitation of 490nm, and collected emission of 515-675 nm. However, neither morphological evidence from microscopic examination nor molecular evidence is provided that the vast majority photoytoplankton in the targeted area is cyanobacteria, esp. Synechococcus.

3 The authors misunderstand the concept of quantum yield and thus the concept was misused. The lifetime of phycobilisome is the lifetime of phycobilisome, not the lifetime of photosynthetic major pigment chlorophyll, nor the quantum yield of photochemistry of PSII (more common symbol Fv/Fm). So when discussing some photosynthetic models, actually there is no data to support the discussion and induced assumptions.

4 Even to calculate the quantum yield of fluorescence lifetime of phycobilisome, the natural lifetime was incorrectly measured. The natural lifetime was calculated based on quantum physics,

not lifetime from isolated phycobilisome particles. This lifetime from isolated phycobilisome is another actual lifetime but in the solvent when isolating and keeping the phycobilisome particles. This measured one is definitely much shorter than the real natural one because the excited energy decay in the solvent interfere the natural decay process.

5 A tiny suggestion when citing the equation of quantum yield of fluorescence: please add more original short paper written by Prof. Brody in 1957 in Science. Also in that paper, a basic feeling of why the natural lifetime was wrongly determined could be get after reading it.

Dear reviewers,

15/04/2022

Enclosed please find our revised manuscript: *COMMSBIO-21-3100-T Title: "Phycobilisome light-harvesting efficiency in natural populations of the marine cyanobacteria Synechococcus increases with depth"*

We thank all three reviewers for their reports, and hereby address all of the reviewer's comments point by point, after changing the manuscript accordingly. Changes in the manuscript are highlighted.

In particular, we address all the concerns raised by reviewer 3 regarding the experimental design and the use of phycobilisome lifetime to estimate light-harvesting efficiency. To address these comments, we performed additional experiments and revised the text to explain more precisely the arguments presented in the manuscript (see new Supplemental figure 3).

Reviewer comments

פרופ' ניר קרן
המחלקה למדעי הצמח והסביבה
02-6585233
nir.ke@mail.huji.ac.il

Reviewer #1 (Changes in the manuscript following reviewer's #1 comments are highlighted in yellow throughout the manuscript):

Prof. Nir Keren
Department of plant and
environmental sciences
+972-2-8585233
nir.ke@mail.huji.ac.il

This is a nice example of the collaboration between oceanography and biophysics bridged by photosynthesis research. In this manuscript, the authors showed that phycobilisome of cyanobacterial cells in deeper water had faster decay of fluorescence, suggesting more efficient energy transfer to reaction center. This is interesting as well as only novel finding of this paper. In that sense, Figure 3 alone is essential. This type of concise paper may not be highly evaluated in some types of journals, but the reviewer considers conciseness itself may not be a problem because the fact that the efficiency of energy transfer can be regulated in response to light environments in open sea is very important from the viewpoint of photosynthesis research as well as of ecology. The following points should be considered to improve the manuscript.

בניין סילברמן
קרית אדמונד י' ספרא, גבעת רם, ירושלים
91904

Silberman Building
Edmond J. Safra campus
Givat Ram, Jerusalem 91904,
Israel

1. Discussion is quite qualitative. Even though this is the first report on the depth-dependency of antenna regulation of cyanobacteria in natural water, there are several reports on this point for cultured cells, including the works of authors

themselves. Please compare the values of fluorescence decay, fluorescence yield or yield of photosynthesis with the values in the past studies of cyanobacteria and other photosynthetic organisms and discuss them more quantitatively.

Reply: The discussion was elaborated to include a comparative part referring to previous reports of cultured cells in laboratory experiments. This paragraph highlights the uniqueness of natural *synechococcus* cultures acclimated to mixed water column conditions as compared to stratified and laboratory culture results. This section can be found in lines 209-226 of the manuscript.

2. For the comparison mentioned above, understanding of the precise experimental condition is essential. The current explanation of the method of Time-Correlated-Single-Photon-Counting refers to the reference 56, but the explanation in the paper is very short and insufficient. The explanation in the reference 19 is more thorough. If the experimental condition is the same, refer to reference 19 and remove reference 56. Furthermore, the fluorescence lifetime must differ at different temperatures. Please specify the temperature during lifetime measurements. And if it is low temperature, please discuss about the physiological relevance of the interpretation.

Reply: The methods section describing time-resolved fluorescence measurements was elaborated. It includes detailed on the Time Correlated Single Photon Counting (TCSPC) measurements performed in this study. This section can be found in lines 330-338. Reference 19 was also added as suggested by the reviewer, however it is important to note that the setup used there also includes the description of temperature dependent measurements, which were not performed in the present research. These measurements are experimentally demanding due to the limitations of the cryogenic systems and require at least 24h for each sample, making them irrelevant for samples acquired at a field study, which must be measured simultaneously for comparison.

3. To calculate absolute quantum yield of fluorescence, the intrinsic lifetime of a phycobilisome complex is essential. For this purpose, the authors prepare the complex biochemically, but it is hard to judge if the obtained intrinsic lifetime is artificially disturbed or not. If possible, it would be nice to have some supporting data that assure the intactness of the complex. Alternatively, it may be possible to strengthen the reliability of the obtained value by comparing it with those reported for similar preparations in the past.

Reply: This point was also raised by reviewer 3. Additional information on the subject can be found in the response to his critique. Values for the intrinsic lifetime are available for chlorophyll. However, we couldn't find previous reports for PBS.

פרופ' ניר קרן
המחלקה למדעי הצמח והסביבה
02-6585233
nir.ke@mail.huji.ac.il

Prof. Nir Keren
Department of plant and
environmental sciences
+972-2-8585233
nir.ke@mail.huji.ac.il

בניין סילברמן
קרית אדמונד י' ספרא, גבעת רם, ירושלים
91904

Silberman Building
Edmond J. Safra campus
Givat Ram, Jerusalem 91904,
Israel

Following the reviewers' comments, we isolated phycobilisomes and estimated their intrinsic lifetime and their fluorescence yield through intensity measurements performed in an integrating sphere spectrometer in addition to the fluorescence lifetime measurements. Information supporting the intactness of the PBS fraction used for the experiments is added as supplementary information (Fig. S3 A).

4. It is interesting to know the "complete picture of the fate of the absorbed energy" but it requires the precise determination of F_v/F_m as well as that of fluorescence yield. In the case of cyanobacteria, determination of correct F_v/F_m is not easy (see e.g. Photosynthesis Research, 133, 63-73). Since the authors seems to use DCMU for the determination of F_m , state transition may not have much interfered with the obtained results. The contribution of phycobilisome and PSI chlorophyll to F_o , however, must have led to underestimation of F_v/F_m . The possible effect of such problem should be discussed.

Reply: We agree and a discussion on the limitation of F_v/F_m measurements in cyanobacteria was added. This section can be found in lines 258-269.

Minor points

5. Please distinguish "pigment" and "chromophore". Linear tetrapyrrole in phycobilin should be chromophore, not pigment, since it covalently binds to phycobiliprotein.

Reply: The term was corrected all along the manuscript following the reviewer's comment (highlighted in the text).

6. Please explain why cyanobacteria used inefficient phycobilisome in upper water, if it is possible to make it efficient by "enhanced coupling between pigments in the phycobilisome". Light absorbance may not be limiting step in photosynthesis in upper water, but, even so, smaller antenna with higher efficiency must be advantageous.

Reply: The following sentence and reference were added to put PBS energy transfer efficiencies in context: "The plasticity of the *Synechococcus* is enabled by the position of the PBS antenna in the inter-thylakoid space. However, at the same time, the intermediate chromophore coupling regime determines energy transfer efficiencies that are considered lower than those of thylakoid membrane internal antenna complexes (ref 60)." This section is highlighted and can be found in lines 76-79 of the manuscript.

פרופ' ניר קרן
המחלקה למדעי הצמח והסביבה
02-6585233
nir.ke@mail.huji.ac.il

Prof. Nir Keren
Department of plant and
environmental sciences
+972-2-8585233
nir.ke@mail.huji.ac.il

בניין סילברמן
קרית אדמונד י' ספרא, גבעת רם, ירושלים
91904

Silberman Building
Edmond J. Safra campus
Givat Ram, Jerusalem 91904,
Israel

7. First line of the last paragraph of the Introduction section: "in in" -> "in"

Reply: Corrected in the re-submitted version and highlighted in the text.

8. First paragraph of the Results section: Please first spell out "CTD" or explain it.

Reply: Corrected in the re-submitted version and highlighted in the text.

9. Second paragraph of the Results section: "chlorophyll increased with depth, indicating a higher concentration of photosynthetic units in cells inhabiting deeper water" is based on the fact that "chlorophyll" is estimated as "per cell" when determined by flow cytometry measurements. This may not be obvious for broad readers of this manuscript.

Reply: Corrected in the re-submitted version and highlighted in the text.

10. Forth paragraph of the Discussion section: "(Kolodny et al., 2020b)" should be reference 19.

פרופ' ניר קרן
המחלקה למדעי הצמח והסביבה
02-6585233
nir.ke@mail.huji.ac.il

Reply: Corrected in the re-submitted version and highlighted in the text.

Prof. Nir Keren

Department of plant and
environmental sciences
+972-2-8585233
nir.ke@mail.huji.ac.il

11. Fifth paragraph of the Discussion section: "mechanism that reacts to the radiation regime" -> "mechanism that respond to the radiation regime"?

Reply: Corrected in the re-submitted version and highlighted in the text.

12. Fifth paragraph of the Discussion section: "possible due to the higher density of phycobilisome rods" -> "possibly due to the higher density of phycobilisome rods"?

בניין סילברמן
קרית אדמונד י' ספרא, גבעת רם, ירושלים
91904

Reply: Corrected in the re-submitted version and highlighted in the text.

Silberman Building
Edmond J. Safra campus
Givat Ram, Jerusalem 91904,
Israel

13. Figures 1,2,S1: It may be better to connect symbols of the same season with lines. In the present figures, some symbols are overlapped with one another, and depth-dependency is not so clear.

Reply: Corrected in the re-submitted version.

14. Figure 3: Do not use arbitrary axis for temperature (X axis) for different panels. It makes the comparison between the panels difficult.

Reply: We tried to follow the reviewer's suggestion, however setting the axis range according to the temperature axis hinders the visibility of the lifetime data. Since the temperature is the secondary factor shown in this figure, while the main factor presented is the lifetime data, we prefer to fix the lifetime axis for easier comparison. To ease the comparison between temperatures, we added labels of temperatures on the lower x-axis for the convenience of the readers.

15. Figure S2: Why the measuring depth is different for different seasons? Please add some explanation in the text.

Reply: The methods section describing the sampling depths was elaborated: "Depths were chosen in order to capture the different states of the water column during summer: the stratified layer, the DCM, and below the DCM. Same depths were followed during winter mixing. An additional depth of 25 m was added during two cruises in October-November, which further characterized the shallower communities. During July and August, samples for flow cytometry were collected from two depths only, that represented stratified shallow 5m and the DCM." This section is highlighted and can be found in lines 304-309 of the manuscript.

פרופ' ניר קרון
המחלקה למדעי הצמח והסביבה
02-6585233
nir.ke@mail.huji.ac.il

Prof. Nir Keren

Department of plant and
environmental sciences
+972-2-8585233
nir.ke@mail.huji.ac.il

Reviewer #2 (Changes in the manuscript following reviewer's #2 comments are highlighted in green throughout the manuscript):

The manuscript of Kolodny et al. presents novel and interesting data on the photophysiological characteristics of in-situ cyanobacterial (*Synechococcus*) populations in the ocean. The study builds on previous culture work by the same group (Kolodny et al. 2020 FEBS J), and effectively amounts to an in-situ confirmation of the key result in this previous work, namely that *Synechococcus* exhibits a decreased fluorescence lifetime within the phycobilisome and hence apparently an associated increased transfer efficiency under decreased light conditions.

In general the manuscript is well written and presented and the results are likely to be of interest, although predominantly probably within the field rather than of

בניין סילברמן
קרית אדמונד י' ספרא, גבעת רם, ירושלים
91904

Silberman Building
Edmond J. Safra campus
Givat Ram, Jerusalem 91904,
Israel

wider interest. There were some aspects of the work which I would like to see improved before recommending publication. In particular that were a number of areas where more detail is required in the methods. This is particularly important as the TCSPC measurements are still some of the first reported from natural oceanic populations of phytoplankton and, to my knowledge, the first depth profiles. There are also a few places in the results where I think the authors have reported information in error.

These and other more minor points are outlined further below.

Specific comments:

Line 41: I don't follow this. The measured lifetimes are shorter at depth, so shouldn't the fluorescence quantum yield be smaller (i.e. see equation line 97)? I think this is probably a typo and the authors have these values the wrong way around? i.e. the fluorescence quantum yield is 18% at 70m and 3.5% at 130m? See also Line 224 (and associated comment below).

Reply: We thank the reviewer for detecting this important "typo". The numbers were indeed flipped by mistake. It was corrected both in the abstract and in the discussion. This section can be found in lines 41-42, 185 of the manuscript (highlighted in the text).

פרופ' ניר קרן
המחלקה למדעי הצמח והסביבה
02-6585233
nir.ke@mail.huji.ac.il

Line 50: It is perhaps worth noting that Prochlorococcus (which is numerically the dominant prokaryotic primary producer in many open ocean systems) does not have a phycobilisome. As a broader contextual question, presumably the phenomenon described is specific to the phycobilisome?

Reply: Following the reviewer's suggestion this comment was added to the introduction and the difference between the antenna systems of these strains is now elaborated: "Prochlorococcus however, use membrane internal light harvesting systems." This section is highlighted and can be found in line 57 of the manuscript.

Prof. Nir Keren
Department of plant and
environmental sciences
+972-2-8585233
nir.ke@mail.huji.ac.il

Line 57: '..light harvesting in the deeper ocean'

Reply: Corrected in the re-submitted version and highlighted in the text.

בניין סילברמן
קרית אדמונד י' ספרא, גבעת רם, ירושלים
91904

Line 79: '... when grown under lower light.'

Reply: Corrected in the re-submitted version and highlighted in the text.

Silberman Building
Edmond J. Safra campus
Givat Ram, Jerusalem 91904,
Israel

Line 150: '...phytoplankton cells are...'

Reply: Corrected in the re-submitted version and highlighted in the text.

Line 174: decapitalize 'The'

Reply: Corrected in the re-submitted version and highlighted in the text.

Line 203: Either 'must travel a longer distance' or 'must travel longer distances'

Reply: Corrected in the re-submitted version and highlighted in the text.

Line 210: 'fit with our previous laboratory study'

Reply: Corrected in the re-submitted version and highlighted in the text.

Line 224: See previous comment. Taking a value of the intrinsic lifetime of around 2.1 I calculate fluorescence quantum yields of 18% and 3.3% at 70 and 130m respectively.

Reply: Corrected in the re-submitted version and highlighted in the text.

פרופ' ניר קרן
המחלקה למדעי הצמח והסביבה
02-6585233
nir.ke@mail.huji.ac.il

Lines 225-226: It is unfortunate if, as appears, there is no direct measurement of Fv/Fm from the in situ samples? I would be cautious in the use of these culture data in comparison to the in situ measurements. At the very least, all of the potential caveats in this calculation have to be clearly outlined to the reader. For example, nutrient status will likely be different for the insitu population versus the culture and may have an influence.

Prof. Nir Keren

Department of plant and
environmental sciences
+972-2-8585233
nir.ke@mail.huji.ac.il

Reply: Indeed. The revised manuscript now includes a more detailed description of the caveats of using Fv/Fm in cyanobacteria (see also response to reviewer #1 highlighted in yellow. This section can be found in lines 258-269 of the manuscript).

Line 247: Not sure this statement can be fully defended as prior measurements of Fv/Fm with depth already indicated that photochemical efficiencies increased at lower light intensities? Also, Line 248, do these models actually make explicit assumptions about the photochemical (or transfer) efficiencies?

Reply: As mentioned in the previous comment, the paragraph was rewritten to present the point with more care.

בניין סילברמן
קרית אדמונד י' ספרא, גבעת רם, ירושלים
91904

Silberman Building
Edmond J. Safra campus
Givat Ram, Jerusalem 91904,
Israel

Line 267: pre-concentration step. Did you measure F_v/F_m before and after pre-concentration in order to establish whether there was any change? If not did you have any other way of establishing whether the pre-concentration step might have influenced the measurements?

Reply: We did not perform F_v/F_m measurements on these samples, but only lifetime measurement. One of the main strengths of lifetime measurements is that unlike many other spectroscopic methods (e.g. fluorescence intensity, absorption, F_v/F_m) they are invariant to concentration. This is now stressed in lines 124-126 of the manuscript: "TCSPC has a distinct advantage over fluorescence intensity-based methods, measuring F_v/F_m for example, as it is does not depend on concentration of the measured sample". This section is highlighted in the text.

Line 269: how were the samples stored in the dark? E.g. what temperature?

Reply: The samples were stored in a dark vial, as now stated in the methods section: "Fluorescence lifetime was measured after dark adaptation of 4 h, during which the samples were kept at room temperature". This section can be found in line 313 of the manuscript (highlighted in the text).

Line 283: assume '574/26' is peak and half bandwidth of emission band? Please state this.

פרופ' ניר קרן
המחלקה למדעי הצמח והסביבה
02-6585233
nir.ke@mail.huji.ac.il

Reply: Corrected in the re-submitted version and highlighted in the text.

Line 287: suggest 'excitation was performed at 490 nm'

Reply: Corrected in the re-submitted version and highlighted in the text.

Prof. Nir Keren

Department of plant and
environmental sciences
+972-2-8585233
nir.ke@mail.huji.ac.il

Line 288: was all the signal between 515-675 nm averaged? As indicated above, I would like to see more example data presented, including emission spectra if available and decay curves (see e.g. Figure 4b & c in the cited reference 56).

Reply: Yes, the signal in this optical window is averaged. Most of the fluorescence emission in this range is centered around 575 nm, hence it dominates the TCSPC results. This range was chosen to include all the PBS fluorescence, and exclude the fluorescence of the photosystems (above 675 nm). Examples of raw data graphs were added to the supplementary file (figure S9). Emission spectra is also available, and an example graph was also added to the supplementary (figure S10).

בניין סילברמן
קרית אדמונד י' ספרא, גבעת רם, ירושלים
91904

Silberman Building
Edmond J. Safra campus
Givat Ram, Jerusalem 91904,
Israel

Line 289: define 'IRF' (i.e. Instrument Response Function)

Reply: Corrected in the re-submitted version and highlighted in the text.

Figure 1 caption: 'two representative depths...'

Reply: Corrected in the re-submitted version and highlighted in the text.

Reviewer #3 (Changes in the manuscript following reviewer's #3 comments are highlighted in turquoise throughout the manuscript):

1. The first fatal problem of this paper is the simple experiment design doesn't not support the proposed generalized photosynthetic energy usage model. Only fluorescence lifetime data was collected that is related photosynthesis, and even this lifetime was set for measure phycobilisome lifetime, not the photosynthesis.

Reply: The primary purpose of this research is to study PBS light harvesting machinery, and its changes in response to depth. To make this point absolutely clear we added the word phycobilisome to the title for emphasis:
"Phycobilisome light-harvesting efficiency in natural populations of the marine cyanobacteria *Synechococcus* increases with depth" (highlighted in the re-submitted version).

The suggested possible conclusions of our observations, regarding the complete photosynthetic process, are presented in the discussion section. Based on the advice of the reviewers, that section was rewritten to present both the advantages and shortcomings of these calculations (These paragraphs can be found in lines 258-269 of the manuscript, and are highlighted in yellow in the text). Even if there are uncertainties regarding these conclusions, we do not consider them as fatal for the paper, but merely disagreements which do not extend beyond the scope of the discussion.

As for the point raised regarding the fact that only fluorescence lifetime was collected in our study, it is explained and repeated several times that to obtain a full picture of the fate of absorbed energy, all three pathways (fluorescence, heat dissipation and photochemistry) should be measured. However, such measurements have their limitation in field studies. This is exactly why, in this manuscript, we present a new way to estimate the so far "missing" pathway (fluorescence, using TCSCP method). Photoacoustic cannot be applied to field studies, so heat dissipation cannot be measured. Photochemistry is often estimated in oceanography studies using Fv/Fm, which, as stated by reviewers 1 and 2, is problematic - specifically in the case of cyanobacteria. Hence, for the description of how fluorescence lifetime can complement existing measures, we rely on Fv/Fm

פרופ' ניר קרן
המחלקה למדעי הצמח והסביבה
02-6585233
nir.ke@mail.huji.ac.il

Prof. Nir Keren

Department of plant and
environmental sciences
+972-2-8585233
nir.ke@mail.huji.ac.il

בניין סילברמן
קרית אדמונד י' ספרא, גבעת רם, ירושלים
91904

Silberman Building
Edmond J. Safra campus
Givat Ram, Jerusalem 91904,
Israel

values from *Synechococcus* in a laboratory setting. Using DCMU allows to partially overcome the inherent limitation of PAM or FRRF measurements of cyanobacteria. The energy model is only shown to explain how this new method can contribute to calculations of the photosynthetic efficiency. To clarify this point, we rewrote that section of the discussion.

As for the correlation between fluorescence lifetime of the phycobilisome and the complete photosynthetic system:

1. Light-harvesting (performed by PBS) is the first step of the photosynthetic process, and its efficiency is an integral part of the complete process. Modifications in its efficiency directly influence the overall efficiency of the process. Previously, we measured both PBS fluorescence lifetime and the photochemistry quantum yield, using several methods, and showed that they are indeed correlated [Kolodny et al 2019, FEBS journal].

2. Measuring fluorescence lifetime of PBS and not of chlorophylls (i.e. the photosystems) was done intentionally, as this allows experimentally to target only *Synechococcus*, without measuring the diverse range of the photosynthetic organisms which exist in the seawater samples.

3. This section was rewritten, following the reviewer's comment, clarifying the extent to which this approach can be applied to natural systems (lines 258-269 of the manuscript).

2. In the study, fluorescence lifetime from the water sample were purposefully set at the excitation of 490nm, and collected emission of 515-675 nm. However, neither morphological evidence from microscopic examination nor molecular evidence is provided that the vast majority phytoplankton in the targeted area is cyanobacteria, esp. *Synechococcus*.

Reply: Several types of evidence are provided, and this point is addressed in the text (lines 128-132): "...excitation at 490 nm, directed at the excitation peak of PE; Emission between 515-675 nm, capturing only phycobilisome fluorescence and minimizing the contribution of chlorophylls. Chryptophyceae, also contain PE. Yet, these are considerably larger than *Synechococcus* 40 and were removed from the samples by size-filtration. Their concentrations were negligible, as validated by flow cytometry." Size filtration was used to remove Chryptophyceae from the samples. Most importantly, the FACS measurements in this research were conducted specifically to address this point, so that we won't be "blinded" as to what is the content of the seawater samples, and to validate what is being optically measured.

3. The authors misunderstand the concept of quantum yield and thus the concept was misused. The lifetime of phycobilisome is the lifetime of phycobilisome, not the lifetime of photosynthetic major pigment chlorophyll, nor the quantum yield

פרופ' ניר קרן
המחלקה למדעי הצמח והסביבה
02-6585233
nir.ke@mail.huji.ac.il

Prof. Nir Keren
Department of plant and
environmental sciences
+972-2-8585233
nir.ke@mail.huji.ac.il

בניין סילברמן
קרית אדמונד י' ספרא, גבעת רם, ירושלים
91904

Silberman Building
Edmond J. Safra campus
Givat Ram, Jerusalem 91904,
Israel

of photochemistry of PSII (more common symbol Fv/Fm). So when discussing some photosynthetic models, actually there is no data to support the discussion and induced assumptions.

Reply: The concern regarding the distinction between lifetime of PBS and lifetime of chlorophyll was raised in the first comment by the reviewer, and as explained in our answer they are indeed not the same, but related to each other. lifetime of PBS is indeed not the QY of photochemistry of PSII. We specifically state the linear relation between lifetime and QY of fluorescence, and the inverse relation between QY of fluorescence and QY of photochemistry of PSII, both in the introduction (lines 97-104) and in the discussion (lines 258-269).

4. Even to calculate the quantum yield of fluorescence lifetime of phycobilisome, the natural lifetime was incorrectly measured. The natural lifetime was calculated based on quantum physics, not lifetime from isolated phycobilisome particles. This lifetime from isolated phycobilisome is another actual lifetime but in the solvent when isolating and keeping the phycobilisome particles. This measured one is definitely much shorter than the real natural one because the excited energy decay in the solvent interfere the natural decay process.

Reply: We thank the reviewer for this comment. To resolve this issue, we isolated *Synechococcus* phycobilisomes and measured the fluorescence quantum yield using an integrating sphere fluorimeter setup and then used these values to calculate the natural lifetime of PBS fluorescence. This section can be found in lines 174-190 of the manuscript. To the extent of our knowledge this is the first time this value is reported. The results are detailed in figure S3.

5. A tiny suggestion when citing the equation of quantum yield of fluorescence: please add more original short paper written by Prof. Brody in 1957 in Science. Also in that paper, a basic feeling of why the natural lifetime was wrongly determined could be get after reading it.

Reply: We thank the reviewer for this suggestion, we cited this paper and performed an additional measurement as done in this research (see answer to comment 4). This section can be found in lines 178-180 of the manuscript: "Following the methodology presented by Brody and Rabinowitch for chlorophyll, we first evaluated the fluorescence quantum yield Φ_f through intensity measurements using an integrating sphere spectrometer". This section was highlighted in the text.

פרופ' ניר קרון
המחלקה למדעי הצמח והסביבה
02-6585233
nir.ke@mail.huji.ac.il

Prof. Nir Keren
Department of plant and
environmental sciences
+972-2-8585233
nir.ke@mail.huji.ac.il

בניין סילברמן
קרית אדמונד י' ספרא, גבעת רם, ירושלים
91904

Silberman Building
Edmond J. Safra campus
Givat Ram, Jerusalem 91904,
Israel

The Alexander Silberman
Institute of Life Sciences
The Hebrew University of Jerusalem

המכון למדעי החיים
ע"ש אלכסנדר סילברמן
האוניברסיטה העברית בירושלים

We thank the reviewers for their valuable comments. A revised version of the manuscript according to their suggestions was uploaded with changes addressed by color code to each reviewer.

We are confident that the corrected manuscript in its present form will be appropriate for publication in Communications Biology Journal.

With best regards,

In behalf of the authors,

Prof. Yossi Paltiel

Prof. Nir Keren

פרופ' ניר קרן
המחלקה למדעי הצמח והסביבה
02-6585233
nir.ke@mail.huji.ac.il

Prof. Nir Keren

Department of plant and
environmental sciences
+972-2-8585233
nir.ke@mail.huji.ac.il

בניין סילברמן
קרית אדמונד י' ספרא, גבעת רם, ירושלים
91904

Silberman Building
Edmond J. Safra campus
Givat Ram, Jerusalem 91904,
Israel

REVIEWERS' COMMENTS:

Reviewer #1 (Remarks to the Author):

In this revised manuscript, the authors satisfactorily answered to the points raised by the reviewer. There are several very minor points that should be mentioned.

1. As for Time Correlated Single Photon Counting (TCSPC) technique, the authors seem to misunderstand the reviewer's comment. The reviewer did not intend to request the measurement at low temperature. Since the measurements were carried out in the past at low temperature as well, the measurements temperature should be specified. Please simply put "at room temperature" in the description of the methods (line 331) for clarity.
2. line 253: "Phycourubilin" -> "Phycourobilin"
3. line 327: "Synechococcus" -> "Synechococcus"
4. line 342: "PSI ,twice" -> "PSI, twice"
5. line 342: "1150g" -> "1150 g"
6. There are "photo acclimation" and "photo-acclimation" in the text. "photo-acclimation" may be better.

Response to Referees

Reviewer #1:

In this revised manuscript, the authors satisfactorily answered to the points raised by the reviewer. There are several very minor points that should be mentioned.

1. As for Time Correlated Single Photon Counting (TCSPC) technique, the authors seem to misunderstand the reviewer's comment. The reviewer did not intend to request the measurement at low temperature. Since the measurements were carried out in the past at low temperature as well, the measurements temperature should be specified. Please simply put "at room temperature" in the description of the methods (line 331) for clarity.

Reply: Corrected in the manuscript following the reviewer's comment.

2. line 253: "Phycourubilin" -> "Phycourobilin"

Reply: Corrected in the manuscript.

3. line 327: "Synechococcus" -> "Synechococcus"

Reply: Corrected in the manuscript.

4. line 342: "PSI ,twice" -> "PSI, twice"

Reply: Corrected in the manuscript.

5. line 342: "1150g" -> "1150 g"

Reply: Corrected in the manuscript.

6. There are "photo acclimation" and "photo-acclimation" in the text. "photo-acclimation" may be better.

Reply: The term was corrected throughout the manuscript following the reviewer's comment.